# Time in Range for Closed-Loop Systems versus Standard of Care during Physical Exercise in People with Type 1 Diabetes: A Systematic Review and Meta-Analysis

**DOI:** 10.3390/jcm10112445

**Published:** 2021-05-31

**Authors:** Max L. Eckstein, Benjamin Weilguni, Martin Tauschmann, Rebecca T. Zimmer, Faisal Aziz, Harald Sourij, Othmar Moser

**Affiliations:** 1Division of Exercise Physiology and Metabolism, Department of Sport Science, University of Bayreuth, 95440 Bayreuth, Germany; max.eckstein@uni-bayreuth.de (M.L.E.); rebecca.zimmer@uni-bayreuth.de (R.T.Z.); 2Interdisciplinary Metabolic Medicine, Division of Endocrinology and Diabetology, Department of Internal Medicine, Medical University of Graz, 8036 Graz, Austria; benjamin.weilguni@medunigraz.at (B.W.); faisal.aziz@stud.medunigraz.at (F.A.); ha.sourij@medunigraz.at (H.S.); 3Department of Pediatrics and Adolescent Medicine, Medical University of Vienna, 1090 Vienna, Austria; m.tauschmann@meduniwien.ac.at

**Keywords:** artificial pancreas, time in range, type 1 diabetes, exercise

## Abstract

The aim of this systematic review and meta-analysis was to compare time in range (TIR) (70–180 mg/dL (3.9–10.0 mmol/L)) between fully closed-loop systems (CLS) and standard of care (including hybrid systems) during physical exercise in people with type 1 diabetes (T1D). A systematic literature search was conducted in EMBASE, PubMed, Cochrane Central Register of Controlled Trials, and ISI Web of Science from January 1950 until January 2020. Randomized controlled trials including studies with different CLS were compared against standard of care in people with T1D. The meta-analysis was performed using the random effects model and restricted maximum likelihood estimation method. Six randomized controlled trials involving 153 participants with T1D of all age groups were included. Due to crossover test designs, studies were included repeatedly (a–d) if CLS or physical exercise interventions were different. Applying this methodology increased the comparisons to a total number of 266 participants. TIR was higher with an absolute mean difference (AMD) of 6.18%, 95% CI: 1.99 to 10.38% in favor of CLS. In a subgroup analysis, the AMD was 9.46%, 95% CI: 2.48% to 16.45% in children and adolescents while the AMD for adults was 1.07% 95% CI: −0.81% to 2.96% in favor of CLS. In this systematic review and meta-analysis CLS moderately improved TIR in comparison to standard of care during physical exercise in people with T1D. This effect was particularly pronounced for children and adolescents showing that the use of CLS improved TIR significantly compared to standard of care.

## 1. Introduction

Achieving glycemic targets remains a major hurdle in people with type 1 diabetes (T1D). Less than 40% of patients have a so called “optimal glycemic control” despite recent advancements in glucose sensing technologies, insulin pumps, and insulins [1]. A recently published position statement highlighted advancements in diabetes technology around physical exercise to support both health care professionals (HCP) and people living with T1D in their therapy management [2]. This is not only of interest for adults with T1D, but also for children and adolescents. Physical activity and exercise are also recommended to be performed as often as possible since the health benefits of being physically active are well established for this young population [3,4,5]. 

Closed-loop systems (CLS), also called “artificial pancreas” systems, may facilitate diabetes management around physical exercise. These systems are able to automatically adjust exogenous insulin administration based on sensor glucose levels in line with dosing algorithms [6,7]. Those systems have already proven to be safe and efficient in several randomized controlled trials in adults [8], children and adolescents [9], during day- and nighttime [10,11], and in inpatient [12] and outpatient settings [13]. Additionally, CLS performed well during glycemic challenges such as meals or unannounced physical exercise [14]. The safety and efficacy under environmental conditions have also been confirmed in a recent meta-analysis, comparing CLS against standard of care, leading to 15.15% (95% CI 12.21% to 18.09%) more time spent in range (70–180 mg/dL (3.9–10.0 mmol/L)) when using a CLS [15].

In general, physical exercise is one of the biggest challenges for glycemic management in people with T1D [16]. Glucose levels may increase, stay stable, or decrease depending upon exercise type, duration, intensity, and experience [2]. In particular, children and adolescents require support during physical exercise since they are more prone to dysglycemia resulting in elevated risk of hyper- and hypoglycemia, especially during post-exercise nocturnal periods [17]. 

CLS have the potency for participation in physical exercise by self-learning algorithms to withhold or deliver micro-doses of insulin to maintain euglycemia. Yet, those systems rely on continuous glucose monitoring (CGM) technology, which has been shown to be less accurate during physical exercise and rapid glucose changes, where mean absolute relative differences of up to 26.5% were observed [18,19]. The physiological lag-time between interstitial fluid, in which the glucose sensor is placed, and capillary or venous blood remains the biggest challenge for CLS algorithms [20]. Furthermore, since exogenous insulin kinetics are only mimicking endogenous glucose lowering effects, CLS may struggle with timely reactions to hypo- and hyperglycemia, especially during physical exercise [21]. Even though faster acting bolus insulins are available now, two main differences to endogenous insulin are present: no first-pass through portal circulation and their subcutaneous absorption is significantly slower in comparison to endogenous insulin [22].

Consequently, it is of interest to assess if CLS are superior or even inferior to standard of care during physical exercise in people with T1D. Therefore, this systematic review and meta-analysis aimed to investigate the percentage of time in range (TIR) of CLS in comparison to standard of care during physical exercise in people with T1D.

## 2. Materials and Methods

This systematic review and meta-analysis is reported in accordance with the PRISMA guidelines and was registered at *the International Prospective Register of Systematic Reviews* (http://www.crd.york.ac.uk/prospero; CRD42929136669, 24th May 2019) [23]. 

### 2.1. Search Strategy and Selection Criteria 

We selected relevant studies published from inception until the 21st of January 2020 by searching EMBASE, PubMed, Cochrane Central Register of Controlled Trials and ISI Web of Science. We used combined terms medical subject headings (MeSH) and text words (tw) to search relevant studies. Search terminology and strategy can be found in the Appendix A). Potentially eligible studies were considered for review if the articles were written in English. Studies were excluded if time in range (70–180 mg/dL (3.9–10.0 mmol/L)) in people with T1D during physical exercise while on CLS or standard of care was not reported. Likewise, we excluded the studies if neither type nor duration of physical activity was defined.

### 2.2. Data Extraction and Quality Assessment

Two independent investigators (M.L.E. and B.W.) reviewed the study titles and abstracts. Studies that satisfied the inclusion criteria were then retrieved for full-text evaluation (Appendix A). Disagreements were resolved by two other investigators (O.M., and H.S.). The following data were extracted from each selected study: study characteristics (author, publication year, journal country, city, continent, type of study), population (pathology, number of participants, age range, participants completed the study, gender, age of participants that completed the study, anthropometrics), outcome measures (device (CGM, pump), type of drug (if applicable), algorithm, time in range, dysglycemic events, number of experiments), and performance parameters (exercise type, duration, intensity, frequency). If data were missing, authors were contacted to receive data. In case of data unavailability, e.g., missing TIR or missing comparator (standard of care) these studies were excluded. 

Studies were independently assessed by two investigators (M.L.E. and B.W.) for methodological quality using the risk of bias assessment (ROB) tool from the Cochrane Collaboration [24] in its revised version [25]. The following sources of bias were detected: overall bias, selection of the reported result, bias of the measurement of the outcome, missing outcome data, deviations from the intended interventions and randomization process (Figure 1). We did not exclude any studies based on the risk of bias assessment since included trials were judged as low risk of bias following the assessment.

### 2.3. Data Synthesis and Analysis 

A narrative descriptive analysis was performed to summarize the characteristics of studies such as year of publication, trial design, sample size, country where the study was conducted, age of the participants, types of interventions, and duration of interventions. TIR was defined as 70–180 mg/dL (3.9–10.0 mmol/L) measured by interstitial CGM devices as described by Battelino et al. [26]. CLS were defined as non-adjustable insulin pump systems (single-hormone) or insulin and glucagon pump systems (dual-hormone) with glucose sensing technology and algorithm-based insulin delivery, which are considered as “fully closed-loop systems” within this systematic review and meta-analysis [27]. Adjustable dual-hormone systems were considered as CLS since sensor-augmented pump dual-hormone systems were not commercially available when performing the search. 

Standard of care is defined as any kind of standard care provided to people with T1D. This also includes predictive low glucose suspend systems (PLGS), open-loop and hybrid CLS, since those systems were made available by the US Food and Drug Administration (FDA) almost two years ago and demand action by the individual if deemed necessary [28]. Where required, means and standard deviations were calculated using appropriate equations (SE = SD√n) [29]. If studies included more than one appropriate data set, these data were extracted and analyzed separately. 

### 2.4. Meta-Analysis

The meta-analysis was performed using the random effects model and restricted maximum likelihood estimation method. The absolute mean difference with corresponding 95% CI was used to summarize the overall and within study difference in TIR between intervention and control groups. The positive mean difference indicated a higher TIR in the intervention group compared with the control group, while the negative mean difference indicated a higher TIR in the control group than the intervention group. The results of the meta-analysis were presented as forest plots. 

## 3. Results

A total of six clinical trials were extracted from 1474 articles that met the objectives [30,31,32,33,34,35]. The steps of the article selection process are described as a flow diagram in Figure 2. Studies published between 2013 and 2019 enrolling a total of 153 participants were included. Due to the crossover designs of four clinical trials, the results have been split for meta-analyses [30,31,32,33]. Therefore, the total population increased to 266 participants and 12 comparisons. Four studies showed a low risk of bias [32,33,34,35], while two studies showed some risk of bias [30,31].

One study was conducted in a crossover trial and compared a single hormone CLS with both predictive low-glucose suspend systems and regular diabetes treatment, both considered as standard of care for this analysis [33]. One study compared a dual-hormone system with and without exercise announcement [31] vs sensor-augmented pump systems, leading to two comparisons. Another study compared single-hormone CLS with standard of care with different exercise types, leading to two comparisons [30]. One study compared a single and a dual-hormone CLS with the standard of care and PLGS, resulting in four comparisons [32].

All other included studies compared single hormone CLS versus standard of care leading to six comparisons [30,33,34,35]. All included studies compared both treatment options during physical exercise at a moderate intensity [31,32,33,34,35], while Dovc et al. also investigated the effects of moderate intensity exercise and sprints [30]. Time under exercise conditions in the studies ranged from 40 min [30] to 330 min [34]. Four studies were conducted under laboratory conditions with an exercise duration of ≤60 min [30,31,32,33]. Two studies were conducted under cold outdoor conditions, which may explain the increased duration of the physical exercise period [34,35]. Two studies conducted the exercise test on a cycle ergometer [30,33], two studies on a treadmill ergometer [31,32], and two studies outside while skiing [34,35]. Four studies were conducted in adolescents of age <18 years [30,33,34,35], while two studies were conducted in adults [31,32]. General study characteristics are summarized in Table 1.

TIR was higher with an absolute mean difference (AMD) of 6.18%, 95% CI: 1.99 to 10.38% in favor of CLS (Figure 3). 

### 3.1. Subgroup Analysis

Subgroup analysis based on age showed the following parameters for studies conducted in adolescents (<18 years): mean difference 9.46% (95% CI −2.48% to 16.45% six study arms, I^2^ = 65%, τ^2^ = 51.43, *p* = 0.01) and for studies conducted in adults (>18 years): mean difference 1.07% (95% CI −0.81% to 2.96%); six study arms, I^2^ = 0%, τ^2^ = 1.15, *p* = 0.44 (Appendix A). For the cycle ergometer the mean difference in time in range was 12.34% (95% CI 4.09 to 20.58%; four study arms, I^2^ = 70%, τ^2^ = 53.58, *p* = 0.02), for the treadmill ergometer the mean difference was 1.07% (95% CI −0.81% to 2.96%; six study arms, I^2^ = 0%, τ^2^ = 1.15, *p* = 0.44) and for skiing the mean difference was 1.35% (95% CI −7.95% to 10.65%; two study arms, I^2^ = 0%, τ^2^ = 0, *p* = 0.09) (Appendix A). In subgroup analysis for study type, a mean difference for crossover randomized trials of 6.87% (95% CI 2.17 to 11.58; I^2^ = 85%, τ^2^ = 39.06, *p* < 0.01, ten studies) and a mean difference of 1.35% (95% CI −7.95% to 10.65%; I^2^ = 0%, τ^2^ = 0.0, *p* = 0.88, two studies) for studies of parallel design was shown (Appendix A).

### 3.2. Secondary Outcomes

The included studies reported only few data regarding our predefined secondary outcomes. The reported data are again given as percentage of total time in the given time period. Due to missing data points and the high variability of the available data, no statistical analyses were performed as very heterogeneous results were to be expected. Detailed data are shown in Appendix A.

Reported data regarding time spent in hypoglycemia (<70 mg/dL; <3.9 mmol/L) during exercise showed values between the lowest of 0.0 (0.0-0.0) (no hypoglycemic event) reported in the intervention group in [30] a, up to a Mean of 8.3 ± 12.6% time in hypoglycemia in the intervention group by Castle et al. [32]. However, due to the lack of data no conclusions can be made whether the results favor the intervention or comparison group.

For five studies data were reported regarding the time spent in hyperglycemia (>180 mg/dL; >10 mmol/L) during exercise, showing large interquartile ranges and variances for all data points. In the intervention group in the study by Dovc et al. [30] a; the time in hyperglycemia was lowest at 17.1% (7.2–33.0) while in the comparative group in the study from Ekhlaspour et al. [35], it was highest at 41.5 ± 30.3% (Appendix A). The number of hypoglycemic events (<70 mg/dL; <3.9 mmol/L) during physical exercise was reported for the study by Dovc et al. [30] b and by Breton et al. [34]. In the study from Dovc et al. [30] b, one hypoglycemic event occurred during physical exercise in the intervention and four in the comparison group. For the study by Breton et al. [34], overall 0.3 ± 0.4 events in the intervention and 0.3 ± 0.7 in the comparison group were reported.

Five included studies reported data for time in range during the post-exercise period. In these studies, the evening or night after exercise was considered as the post-exercise period. Data in all five study arms favored the intervention group, however there were large interquartile ranges and variances. In study [34], time in range during the post-exercise period was 79.3% ± 29.8 in the intervention and 68.8% ± 24.1 in the comparison group. Five included studies reported data for time spent in hypoglycemia (<3.9 mmol/L) during the post-exercise period. All data points are close to 0.0 with the highest values reported in study [34] with 2.2% ± 2.3 in the intervention group and 2.5% ± 6.5 in the comparison group for time in hypoglycemia. 

Three included studies reported data for the time spent in hyperglycemia (>10 mmol/L) during the post-exercise period. In these studies the intervention group showed less time in hyperglycemia with the biggest difference in study [33] a, with 1.0% (0.0 to 13.9%) in the intervention group and 64.6% (1.6 to 97.5%) in the comparison group. 

The number of hypoglycemic events during the post-exercise period was reported in two included studies. In study [30] a, three hypoglycemic events occurred in the intervention and in the comparison group. Breton et al. reported a number of hypoglycemic events in the intervention group with 0.1 ± 0.3 and 0.1 ± 0.4 in the comparison group during the post-exercise period [34].

### 3.3. Assessment of Publication Bias

The publication was assessed in terms of the meta-bias. Specifically, a funnel plot was generated to visualize the publication bias and Egger’s test was performed to assess the asymmetry of the data and the impact of publication bias. The corresponding *p*-values <0.05 would indicate a significant publication bias (Appendix A). The funnel plot indicates moderate asymmetry (*p* = 0.13), suggesting that publication bias cannot be excluded as a factor of influence on the present meta-analysis (Appendix A). It remains possible that due to the lack of studies in this specific research field, studies with smaller samples have not been published or failed to do so.

## 4. Discussion

Our meta-analysis indicated that CLS have superior effects on the TIR during physical exercise in people with T1D when compared against standard of care. Furthermore, in a subgroup analysis for age, CLS were especially favorable in children and adolescents, detailing 9.46% more time spent in range when compared to standard of care. This finding contrasted with the results found in adults where only a 1.07% improvement in TIR was observed. These results may potentially be due to specific alterations in the designs of the studies (e.g., type of exercise or set-up). Intriguingly, one might expect less beneficial effects of CLS in the younger population due to elevated glycemic variability [36] that may affect exogenous insulin requirements and deteriorate CGM accuracy, which is seen during moderate-intensity exercise [2,37,38]. Two recent position statements gave recommendations on how to manage glucose around physical exercise according to values given by intermittently scanned glucose monitoring systems (isCGM) and CGM sensors [2,3]. These statements have also shown a Mean Absolute Real Difference (MARD) of 13.6% over all systems that may deteriorate the performance of CLS. Independent of how good an algorithm is, sensor accuracy is reduced due to the physiological lag time between blood and interstitial glucose during acute physical exercise, and therefore remains the biggest issue in maintaining euglycemia [39]. This is partly supported by our analysis (Appendix A) since both CLS and standard of care lead to comparably lower time in hypoglycemia, whereas time in hyperglycemia is higher during standard of care. However, these findings must be interpreted cautiously since most studies included in our analysis did not show these data.

Physical exercise performed on a cycle ergometer showed beneficial effects of CLS on TIR when compared against standard of care. In contrast, this was not seen when the exercise was performed on a treadmill ergometer or during a skiing session. The metabolic effects of exercise are often generated based on the absolute amount of muscle mass used [40]. Taking this into account and hypothesized in a previous letter [41], treadmill exercise and skiing may yield a greater rate of change in glucose, hence deteriorating CLS performance. 

For most people with T1D, scheduled physical exercise is favorable to spontaneous exercise since insulin dosing can be adjusted prior to exercise and additional carbohydrates can be consumed if necessary. However, the glycemic management in the post-exercise phase has proven to be difficult for people with T1D. Due to increased glucose transporter type -4 activity, glucose uptake is increased for up to 24 h following exercise [16], which may induce hypoglycemia. On the other hand, fear of hypoglycemia and excessive carbohydrate consumption during exercise may lead to hyperglycemia, which deteriorates HbA1c over extended periods of time [42,43]. This may also explain our findings in Appendix A. Time in range appears to be higher in the post-exercise period compared to standard of care; however, we have not conducted formal analysis given that this outcome is available in only a few studies. Notably, time in hypoglycemia appears similar with a maximum of 2.2 ± 2.3% in CLS and 2.5 ± 6.5% during standard of care [34]. Hypoglycemic events were also comparable, with a total of three in CLS and four in standard of care [30] (Appendix A).

A recent study conducted by Viñals et al. investigated the effects of the performance of a multivariable CLS controller with automatic carbohydrate recommendation during and after announced and unannounced physical exercise in people with T1D [44]. It was shown that CLS in both conditions increased the time in range, avoided hypoglycemia and reduced time spent in hyperglycemia during post-exercise conditions [44], which is in line with the findings from our meta-analysis. A previous systematic review and meta-analysis investigated the efficacy and safety of CLS in 2017 [15]. This analysis concluded that the devices are safe and efficacious in treating outpatients with T1D. A more recent network meta-analysis had similar findings by underpinning that CLS led to a lower median glucose of 0.75 mmol/L compared with open-loop systems. They concluded that the DiA platform was superior to other platforms with an advantage for dual-hormone systems to lower mean glucose and increase time in range [45]. Due to the lack of studies applying dual-hormone systems during physical exercise, a comparison within our meta-analysis was not possible.

CLS have raised awareness in the population of people with T1D using continuous subcutaneous insulin infusion systems, which led to the popularity of do-it-yourself CLS that have become present within a small community of people with T1D. However, this still remains an ethical issue for health-care professionals since CLS, even though successfully tested for research purposes over decades, are not yet commercially available [46]. However, the complexity of implementing these devices in glycemic therapy and making them commercially available is reliant on research. The potential of CLS to ameliorate exercise management in T1D has previously been discussed in a narrative review by Bally and Thabit, which is in line with our findings [47] We suspect that due to the velocity at which technological innovations for CLS are released, running research is already outdated prior to publication, which in turn hinders the market release of those devices. 

Our meta-analysis has the same limitations as other previously published meta-analyses as the inconsistency in the outcome reporting does not allow a clear conclusion. A further limitation in the interpretation of the results is the complexity of physical exercise since time, intensity, duration and study setting were influencing the results, which might have led to an increased reporting bias. Therefore, more studies in inpatient and outpatient settings with larger sample sizes are needed to overcome this bias in outcome reporting.

## 5. Conclusions

Our findings highlight that CLS lead to a greater time in range during physical exercise when compared against standard of care in particular in children and adolescents. CLS ameliorates glycemic control during exercise in this cohort, may reduce parental fear due to their unique properties and allow more spontaneous physical activity during childhood [48].

The future of glycemic management in people with T1D might be based upon the development of commercially available CLS. These systems should thus be regarded as a safe and efficacious method to maintain glucose control around physical exercise. 

## Figures and Tables

**Figure 1 jcm-10-02445-f001:**
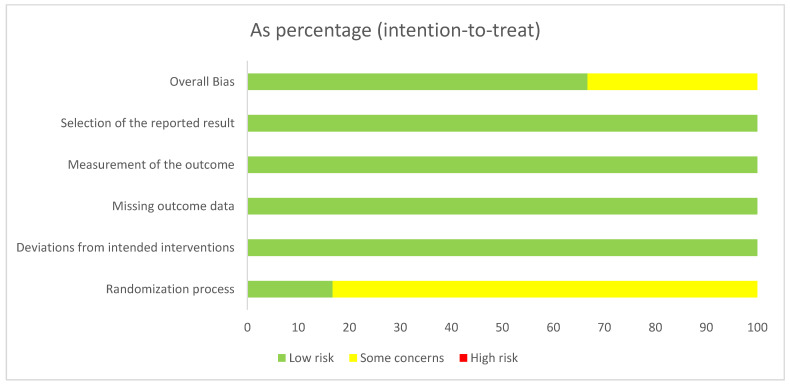
Risk of bias assessment tool. Across trials (n = 6). Information is either from trials at low risk of bias (green), or from trials with some concerns of bias (yellow), or from trials at high risk of bias (red).

**Figure 2 jcm-10-02445-f002:**
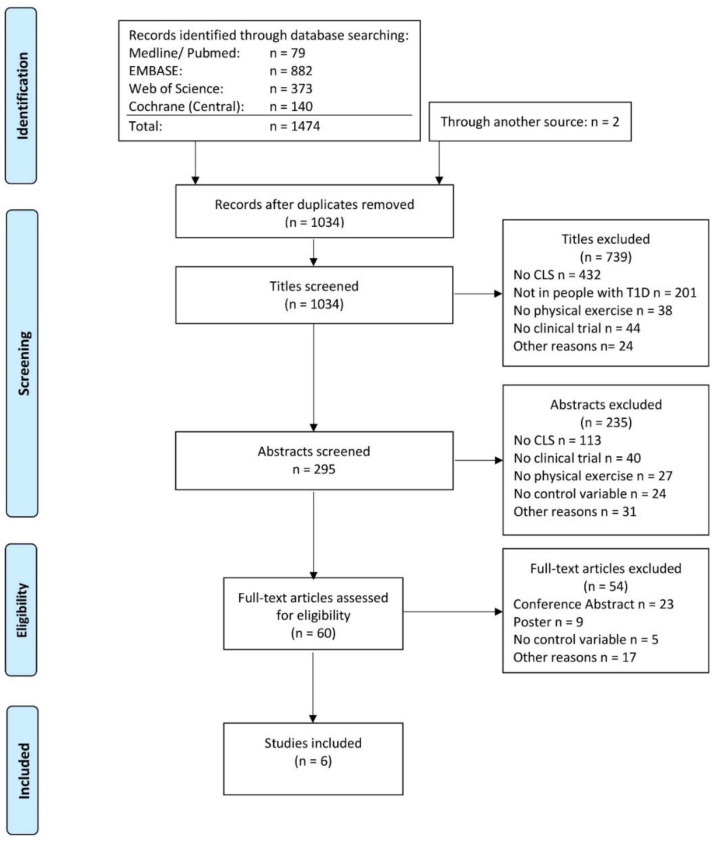
Prisma flow diagram (accessed on, 27th of July 2020).

**Figure 3 jcm-10-02445-f003:**
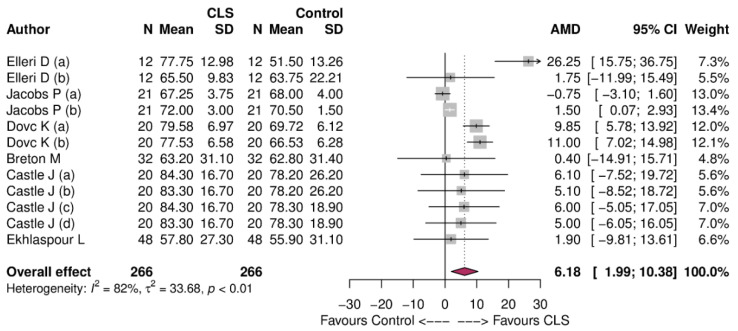
Forest plot of meta-analysis showing the overall effect of control vs. CLS for time in range during physical exercise.

**Table 1 jcm-10-02445-t001:** Generals study characteristics.

	Year	Study Type	n	Age (years)	% TIR CLS	% TIR Control	Device Intervention	Algorithm	Device Control	Exercise Type	Exercise Duration	Exercise intensity
[33] *(a)*	2013	Crossover RCT	12	15.0 ± 1.4	79 (54–99)	56 (24–70)	Animas 2020 Dexcom Seven Plus CGM	model predictive control	Animas 2020 Pump, Dexcom Seven Plus CGM	Cycling	60 min	Moderate-intensity
[33] *(b)*	2013	Crossover RCT	12	15.0 ± 1.4	60 (54–88)	71 (18–95)	Animas 2020 Dexcom Seven Plus CGM	model predictive control	Animas 2020 Pump, Dexcom Seven Plus CGM	Cycling	60 min	Moderate-intensity
[31] *(a)*	2016	Crossover RCT	21	32 ± 7	67 (60–75)	68 (60–76)	Tandem t:slim Dexcom G4 CGM (dual hormone)	Proportional-integral-derivative	Sensor augmented pump; Dexcom G4 CGM	Running	45 min	Moderate-intensity
[31] *(b)*	2016	Crossover RCT	21	32 ± 7	72 (66–78)	68 (60–76)	Tandem t:slim Dexcom G4 CGM (dual hormone)	Proportional-integral-derivative	Sensor augmented pump; Dexcom G4 CGM	Running	45 min	Moderate-intensity
[30] *(a)*	2017	Crossover RCT	20	14.2 ± 2.0	80.9 (64.3–92.2)	68.1 (59.1–83.6)	Paradigm Veo Enlite II sensor	fuzzy-logic	Medtronic Paradigm Veo; Enlite II sensor	Cycling	40 min	Moderate-intensity
[30] *(b)*	2017	Crossover RCT	20	14.2 ± 2.0	75.5 (66.6–92.9)	68.4 (52.1–77.2)	Paradigm Veo Enlite II sensor	fuzzy-logic	Medtronic Paradigm Veo; Enlite II sensor	Cycling	40 min	Moderate-intensity + Sprint
[34]	2017	RCT	32	13.2 ± 1.7	63.2 ± 31.1	62.8 ± 31.4	Tandem t:AP pump/ Roche-Accu-Check Spirit Combo pump Dexcom G4 CGM	DiA	Sensor-augmented pump; Dexcom G4 CGM	Skiing	330 min	Moderate-intensity
[32] *(a)*	2018	Crossover RCT	20	34.5 ± 4.7	84.3 ± 16.7	78.2 ± 26.2	Tandem t:slim pump Dexcom G5 CGM (dual hormone)	Fading memory proportional-derivative	“standard of care”	Running	45 min	Moderate-intensity
[32] *(b)*	2018	Crossover RCT	20	34.5 ± 4.7	83.3 ± 16.7	78.2 ± 26.2	Tandem t:slim pump Dexcom G5 CGM (single hormone)	Fading memory proportional-derivative	“standard of care”	Running	45 min	Moderate-intensity
[32] *(c)*	2018	Crossover RCT	20	34.5 ± 4.7	84.3 ± 16.7	78.3 ± 18.9	Tandem t:slim pump Dexcom G5 CGM (dual hormone)	Fading memory proportional-derivative	Predictive low glucose suspend: t:slim pump, Dexcom G5 CGM	Running	45 min	Moderate-intensity
[32] *(d)*	2018	Crossover RCT	20	34.5 ± 4.7	83.3 ± 16.7	78.3 ± 18.9	Tandem t:slim pump Dexcom G5 CGM (single hormone)	Fading memory proportional-derivative	Predictive low glucose suspend: t:slim pump, Dexcom G5 CGM	Running	45 min	Moderate-intensity
[35]	2019	RCT	48	12.3 ± 3.2	57.8 ± 27.3	55.9 ± 31.1	Tandem t:slimX2 Dexcom G6 CGM	DiA	Sensor-augmented pump; Dexcom G5 CGM	Skiing	240 min	Moderate-intensity

## Data Availability

Data may be obtained from the corresponding author upon reasonable request.

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
