# Peer review of "Time in Range for Closed-Loop Systems versus Standard of Care during Physical Exercise in People with Type 1 Diabetes: A Systematic Review and Meta-Analysis"

_jcm, 2021, doi:10.3390/jcm10112445_

Round 1

Reviewer 1 Report

I applaud the idea of a meta-analysis of this type, but I question the execution of this particular one. Also, given that closed-loop technologies are relatively new, I wonder if it is too early to do so. On a similar note, what is the value of assessing closed-loop only with respect to exercise? If closed-loop systems prove to be of value for overall patient care, does it matter how they perform during exercise? Would the authors advocate not using these systems if during exercise performance is not superior? The more important question regarding these systems is their safety under conditions that stress their capabilities, such as exercise. So I would suggest that looking at safety is a more significant issue than performance.

The authors seek to assess the performance of closed-loop glucose control during exercise versus the standard of care. However, as part of "standard of care," they list hybrid closed-loop systems. Hybrid closed-loop systems ARE closed-loop systems, particularly with respect to exercise. At the same time, they do include two systems that are "hybrid" in the meta-analysis (citations 33 and 34).

I did not carry out a comprehensive search, and I found several papers missing in the meta-analysis, which brings up the question of the criteria used. For instance:

  • Forlenza et al., Performance of Ominipod personalized model predictive control algorithm with moderate intensity exercise in adults with type 1 diabetes, Diabetes Technol Ther, 2019, 21(5):265-272.
  • Huyett et al., Outpatient closed-loop control with unannounced moderate exercise in adolescents using zome model predictive control, Diabetes Technol Ther, 2017, 19(6):331-339.

There is a review article on closed-loop control and exercise, which would be a good reference to include as well. Bally and Thabit, Closing the loop on exercise in type 1 diabetes, Curr Diabetes Rev, 2018, 14(3):257-265.

Some other observations. In the introduction, the authors refer to "...exogenous insulin kinetics are only mimicking endogenous..." While this is correct, it is quite a vague statement. I would point out the main two differences: 1) not having the first-pass through the portal circulation, and 2) the subcutaneous absorption being significantly slower. Actually, for closed-loop as a whole, the biggest obstacle is the slow absorption of subcutaneous insulin (even for the latest faster-acting insulins).

In section 2.1, the last two paragraphs seem to be instructions to the authors.

On line 319, the authors state that closed-loop systems are not commercially available, but this is incorrect. Medtronic's MiniMed 670G, 770G, and 780G systems are all closed-loop. Tandem's Control-IQ is also closed-loop and also commercially available.

In the table listing the studies and the type of algorithm, the data seem to be wrong. For instance, DiA is a made-up name, but this is not the underlying algorithm. Similarly, from what I recall, the dual-hormone systems are model predictive control algorithms, not PID.

Author Response

Dear Reviewer,

thank you very much for reviewing our manuscript. Please find attached our point-by-point response letter.

Kind regards,

Reviewer 2 Report

This systematic review and meta-analysis aimed to investigate the percentage of time in range (TIR) of closed loop systems (CLS) in comparison to standard of care during physical exercise in people with T1D. The findings of the meta-analysis indicated that CLS have moderate superior effects on the TIR during physical exercise in people with T1D when compared against standard of care. In a subgroup analysis for age, CLS were especially favourable in children and adolescents, in terms of more time spent in range when compared to standard of care. The authors acknowledge that the results may potentially be due to specific alterations in the type of exercise or set-up.

Broad comments:

A highly relevant topic is investigated in this paper, contributing with knowledge of clinical importance for diabetes care.

The method is applied according to established standards for systematic review/meta-analysis and the chosen outcome variables are straight-forward and of relevance.

Study limitations are discussed, including assessment of publication bias. 

Conclusion- Why is not the significant findings of greater TIR during physical exercise in children and adolescents included in the conclusion?

Specific comments:

Line 96-98: There is something strange with the wording here. Please reformulate.

Table 1. Could not the papers be presented in chronological order- 2013 (29), 2016 (30) etc?

Figure 3. In line with the meta-analysis approach, a highly relevant and excellent way to present and illustrate the overall findings in a condensed way.

189-190: Why is the results of the subgroup analysis reported here? It comes in the next paragraph.

Line 197: Could Figure S1 be included in the paper instead of in the supplemental material? I do see the point that it is not actually in line with the aim of the article, but it is an interesting finding.

Lind 222 and 225: difficult to see the (29)a and (29)b- could it be clarified that it is referring to the same paper?

Line 230: “evening and night” seem to be of more logical order

Author Response

(The authors gave the same response as above.)
